# Examining Mammalian facial behavior using Facial Action Coding Systems (FACS) and combinatorics

**Aisha Mahmoud[1], Lauren Scott[2], Brittany N. Florkiewicz[3]***

1 Department of Computer and Data Science, Lyon College, Batesville, Arkansas, United States of America,
2 School of Medicine, University of Kansas Medical Center, Kansas, KS, United States of America,
3 Department of Psychology, Lyon College, Batesville, Arkansas, United States of America

* Brittany.florkiewicz@lyon.edu

**Data Availability Statement:** All data underlying the current study will be made available as electronic supplementary material, in addition to

## Abstract

There has been an increased interest in standardized approaches to coding facial movement in mammals. Such approaches include Facial Action Coding Systems (FACS), where individuals are trained to identify discrete facial muscle movements that combine to create a facial configuration. Some studies have utilized FACS to analyze facial signaling, recording the quantity of morphologically distinct facial signals a species can generate. However, it is unclear whether these numbers represent the total number of facial muscle movement combinations (which we refer to as facial configurations) that each species is capable of producing. If unobserved combinations of facial muscle movements are communicative in nature, it is crucial to identify them, as this information is important for testing research hypotheses related to the evolution of complex communication among mammals. Our study aimed to assess how well the existing literature represents the potential range of facial signals in two previously studied species: chimpanzees (*Pan troglodytes*) and domesticated cats (*Felis silvestris catus*). We adhered to the coding guidelines outlined in the FACS manuals, which are based on the anatomical constraints and capabilities of each mammal's face, to create our comprehensive list of all potential facial configurations. Using this approach, we found that chimpanzees and domesticated cats may be capable of producing thousands of facial configurations, many of which have not yet been documented in the existing research literature. It is plausible that some of these facial configurations are communicative and could be discovered with further research and video recording. In addition to our findings having significant implications for future research on the communicative complexity of mammals, it can also assist researchers in evaluating FACS coding accuracy.

## Introduction

For decades, researchers have shown significant interest in studying mammal faces, including the use of facial muscle movements to produce cues and signals [1]. But in recent years, there has been an increasing demand for standardized approaches to identify the various facial

the Python script. There are no restrictions on data availability for this study.

**Funding:** The author(s) received no specific funding for this work.

**Competing interests:** The authors have declared that no competing interests exist.

muscle movements that mammals can produce. One such approach includes the use of Facial Action Coding Systems (FACS). FACS were initially developed for use with humans by Hjortsjö [2], Ekman, and Friesen [3], but were later modified to work with various mammals [4], including apes [5–7], monkeys [8–12], dogs [13], cats [14], and horses [15]. FACS trains individuals to recognize specific facial muscle movements that combine to form facial behaviors, some of which may be communicative (i.e., used as signals). FACS eliminates observer bias by exclusively focusing on the physical form of facial muscle movements, which are all considered equally important [8]. Researchers who want to use FACS must pass a certification test with an expert familiar with the system [4, 16], in addition to conducting inter-observer reliability for their study [15, 17, 18]. With FACS, each facial muscle movement is assigned a unique identification code, which remains consistent across species capable of producing that movement [4]. Consistency in facial muscle movement coding allows for comparisons between species with established FACS.

Research using FACS with non-human mammals has been employed to evaluate communicative facial behaviors, yielding valuable insights into their signaling abilities. For instance, FACS has been utilized to evaluate differences in communication facial configurations and their association with different social interaction outcomes (*Macaca nigra*; [19]), compare the complexity of communication among macaque species with varying degrees of social tolerance [20], assess audience effects during communication in orangutans (*Pongo pygmaeus*; [21], and examine the impact of emotional context on facial signaling in domesticated cats (*Felis silvestris catus*; [22, 23]). FACS have also been used to document the facial signaling repertoires of various mammals, including hylobatids (family Hylobatidae; [17, 24, 25]), chimpanzees (*Pan troglodytes*; [18]), and domesticated cats (*Felis silvestris catus*; [23]). It is important to note that these studies documented facial signaling repertoires mainly to test hypotheses related to the social use of facial movements (among hylobatids; [17, 24]) and to assess differences across species with varying social structures and organizations (*Pan troglodytes* and multiple species of hylobatids; [18, 25]). Although previous ethograms have described the facial signaling abilities of these three mammals [26–29], their classification systems are at risk of overlooking subtle variations in facial muscle movements that could impact the function of these signals [4, 30].

In studies documenting the facial signaling repertoires of hylobatids, chimpanzees, and domesticated cats with FACS, researchers reported the total number of unique facial signals observed (i.e., morphologically distinct facial muscle movement combinations, or configurations, used during bouts of communication). To date, 80 unique facial signals have been identified for hylobatids [18], 357 for chimpanzees [18], and 276 for domesticated cats [23] with the help of FACS. In hylobatids and domesticated cats, there are distinct facial signals (identified with FACS) that are exclusive to affiliative contexts, indicating variations in the social function and use of these muscle movement combinations [23, 25]. Data on the number of unique facial signals identified in hylobatids and chimpanzees has been used to test various hypotheses pertaining to the evolution of facial signaling, such as the socio-ecological complexity hypothesis. One previous study found that chimpanzees produce a greater number of communicative facial configurations compared to hylobatids, and these combinations are also more complex, involving a greater number of discrete facial muscle movements [18]. These data were used to support the idea that socio-ecological factors influence the development of complex facial signaling repertoires in primates, as chimpanzees live in larger social groups compared to hylobatids [31]. Socio-ecological changes during the domestication process have also been suggested as an explanation for why domesticated cats produce a wide variety of unique facial signals [23].

However, it is unclear whether these numbers represent all the unique facial signals that each mammal can generate. When envisioning facial muscle movements as numbers and communicative facial configurations as permutations, the potential facial signaling repertoire of a given species becomes quite extensive. Even when repetition is not allowed, the number of possible facial signals generated is still quite large. For instance, previous research has indicated that the average number of distinct facial muscle movements in a given facial signal is around 3 for chimpanzees, hylobatids, and domesticated cats [18, 23]. If we calculate the number of permutations for a subset of 15 facial muscle movements while discouraging repetition and selecting 3 movements each time, the result is 2,730 possible facial signals. This estimate of the number of morphologically distinct facial signals already exceeds what has been previously documented for all three species [18, 23]. But this estimate may be generous, as it does not consider the relationship between individual facial muscle movements, such as muscle movements that must occur together or are mutually exclusive. Even when implementing these relationships, the range of facial signals may still be extensive. It is possible that challenges related to behavioral sampling have led to smaller documented repertoires.

For instance, 276 unique facial signals have been identified across 3.23 hours of video footage for domesticated cats [23], while 80 have been identified across 227 hours of video footage with hylobatids [18]. It is possible that increasing the amount of video footage for domesticated cats to match that of hylobatids would reveal additional unique facial signals. Increasing the number of study individuals may also reveal additional signals that have yet to be documented. Collecting and coding additional video footage can be time-consuming, potentially limiting research opportunities and insights into the communication behaviors of domesticated cats. For this reason, it is crucial to have a method for assessing whether additional data collection and coding efforts will yield new facial signals or if a limit has been reached. Previous studies have identified communicative plateaus by plotting signaling time (in hours) against the cumulative repertoire size of individuals and species. For instance, studies with chimpanzees (*Pan troglodytes*) have revealed that after 15 hours of active signaling, there are minimal alterations in the gestural repertoire sizes of observed individuals [32].

Using data from 18 individuals, researchers have identified 357 unique facial signals in chimpanzees [18], and 80 combinations across 36 individuals for hylobatids [18]. Increasing the number of chimpanzees sampled could also increase the number of unique facial signals, or if additional signals are not identified, it could suggest a communicative threshold has been reached. Previous studies on the facial signaling repertoires of chimpanzees [18], hylobatids [17, 18, 24, 25], and domesticated cats [23] have primarily focused on adults. When increasing the number of individuals to sample from, it may be beneficial to sample from a diverse range of ages and sexes to create a more representative facial signaling repertoire of the species. But as with hours of video footage, increasing the number of individuals may not necessarily lead to more unique facial configurations, especially if a repertoire maximum has been reached. One analogy that can be used to illustrate the relationship between hours of video footage, number of sampled individuals, and number of identified facial configurations is the sampling efforts associated with obtaining measures of species richness. The greater the sampling effort, the more individuals are collected, and the higher the probability of identifying rare species until a maximum is reached [33]. This could also apply to facial signals: the greater the sampling effort, the more facial signals are collected, and the greater the probability of identifying new facial configurations until a maximum is reached.

Cumulative repertoire plots [32, 34] and discovery curves [33] provide one possible avenue for evaluating facial signal sampling efforts and repertoire thresholds. However, it's important to note that these approaches also have their limitations. Cumulative repertoire plots only display collective values over time, making it challenging to identify sudden fluctuations that may

be due to other sampling factors and obstacles. Identifying and addressing these sampling factors and obstacles could result in a greater variety of facial signals than previously reported. Discovery curves are subject to large margins of error unless the inventory of signals is nearly complete [35]. Furthermore, discovery curves are prone to underestimation and false plateaus [35], which could negatively impact the documentation of facial signaling repertoires. In order to improve the accuracy of cumulative repertoire plots and discovery curves, it is important to use an additional approach that can identify the potential maximum inventory of facial signals and detect any areas of possible sampling bias, such as coding bias.

If only a portion of all possible unique facial signals has been documented for mammals (with established FACS), it is unclear whether these smaller numbers are due to differences in use or visibility. For example, in one previous study, researchers found that hylobatids produce brow movements in their communicative facial signals (such as inner and outer brow raiser, AU1+2), but were absent in chimpanzees [18]. It is possible that differences in video sampling techniques, equipment used for video recording, and/or enclosure design may have led to discrepancies in identifying movements of the brows. Compared to humans, chimpanzees have prominent brows that lack contrasting textures and colors. The morphology of chimpanzee brow ridges may make it difficult to discern brow movements [5], which could explain their low occurrence across multiple studies [18, 36]. It is also possible that brow movements are used for communication in hylobatids but not chimpanzees [18]; however, these movements have been observed in some chimpanzee pant hoot and alert face displays [36]. If chimpanzees communicate with brow movements, it is important to consider their potential impact on the size of the chimpanzee facial signaling repertoire.

## Current study

In our current study, we aimed to assess how well the existing literature represents the possible range of facial signaling among mammals. To achieve this aim, we compiled a comprehensive list of all possible facial muscle movement combinations (referred as facial configurations from this point onward) for mammals with established FACS and previously documented facial signaling repertoires. Although we may not be able to ascertain the communicative function of every facial configuration without additional research, compiling a thorough list of all possible facial configurations can still help in assessing the facial mobility and communication potential in mammals. This information can also be utilized to guide future data collection and coding efforts for each species. The results of our study are not meant to discourage researchers from conducting additional studies, even if millions of facial configurations are identified. Rather, we hope that our study can help researchers identify underexplored facial muscle movements and configurations, offering deeper insights into a species' socio-ecology. We focused on chimpanzees (*Pan troglodytes*; [18]) and domesticated cats (*Felis silvestris catus;* [23]), as their facial signals have been recently documented and coded with their respective FACS. It is worth noting that multiple studies have extensively documented, coded (with FACS), and compared the facial signals of hylobatids [17, 18, 24, 25]. In contrast, chimpanzees [18] and domesticated cats [23] have only one study published that contains information on observed signals. For this reason, we chose to omit hylobatids from our current study and concentrate on underrepresented species.

## Materials and methods

This study involved the creation and comparison of two dataset types. The first type includes "observed" configurations of facial muscle movements for domesticated cats and chimpanzees, while the second type includes all "possible" configurations of facial muscle movements for

both species. To create our "observed" configuration lists for chimpanzees and domesticated cats, we utilized data from previously published studies [18, 23], while focusing solely on facial muscle movements described in each species' respective FACS manuals [14, 36]. These data are only a part of our current study's focus, used for comparison purposes. A brief overview of associated data collection and coding procedures can be found below. Additional information about how these data were obtained and coded can be found in their respective publications [18, 23]. The data for each study can be accessed from the electronic supplement available on the website of the respective journal.

We created our "possible" configuration lists using a combinatorial approach. Specifically, we generated a set of guidelines for each type of facial muscle movement based on FACS manuals and converted them into a Python model. This model then produces a comprehensive list for each species according to those guidelines. Additional information about our combinatorial approach and Python code can be found below.

### Observed combinations—Data collection

Data for our "observed" configuration lists were collected through video recordings in two different locations. Video footage with chimpanzees was collected from 2017–2019 using a combination of the focal [37] and opportunistic sampling method [38] at the Los Angeles Zoo and Botanical Gardens, Los Angeles, California, USA. Video footage with domesticated cats was collected from 2021–2022 using the opportunistic sampling method at the CatCafé Lounge, Los Angeles, California, USA. Data from both locations was conducted in according with the Association for the Study of Animal Behaviour's guidelines for the treatment of animals in behavior research [39]. Our study was approved by the Los Angeles Zoo and CatCafé Lounge before collecting data. Since we used non-invasive behavioral observations to collect our data, our study was exempt from IACUC review. Videos were recorded with a Panasonic Full HD Video Camera Camcorder HC-V770 at both locations. Using these data collection protocols, 156.5 hours of video footage were collected with chimpanzees [18], and 3.23 hours were collected with domesticated cats [23]. Because there are difference in the number of recording hours for both species, we cannot directly compare the observed configuration lists, as the identification rates would not be the same.

### Observed combinations—Facial Action Coding

The data we used to create our "observed" configuration primarily focused on facial signals. In both studies, facial signals are defined as facial muscle movements performed by a signaler when communicating with recipients to elicit a behavioral response [18, 23]. Like other types of communicative signals, facial signals evolved to convey information from signals to recipients and the recipient's behavioral response has positive fitness consequences for both signalers and recipients [40]. For both studies, facial signals that led to a behavioral response or indicated a desired behavioral response were coded. Facial muscle movements and facial configurations produced for biological maintenance (such as chewing) were not considered in these two studies. All identified facial signals were coded using each mammal's respective Facial Action Coding System (chimpFACS; [5, 14]). Facial signals were coded at their apex, or the peak production, when all facial muscle movements involved in the signaling bout were present and at their highest intensity. For example, a mammal may initially produce a facial signal of low intensity, involving only 2 facial muscle movements. A couple of seconds later, an additional 3 muscle movements might be added during the facial signaling bout. It is when all 5 facial muscle movements are present and produced that the apex is reached, and this is when we code the facial signal using FACS. Following the language of the chimpFACS and catFACS,

each facial muscle movement was assigned a unique identification code, and the combination of unique identification codes constitutes a facial signal). A list of all facial muscle movements that were coded and used in our present study for chimpanzees and domesticated cats can be found in the electronic supplement (**S1 Table in S1 File**).

There are different types of facial muscle movements recognized in the FACS. For our current study, we focus only on Action Units (AUs) and Action Descriptors (ADs). Our chimpanzee dataset contained additional movements such as head (M51-M55) and eye (M69) movements, vocalizations (AD50), positional behaviors (S101), manual gestures (S100), and gross behaviors (G84 & G85; [18]), but these were not coded for domesticated cats [23]. One instance of asymmetrical ear movement was identified in the domestic cat dataset [23], but asymmetrical coding was not implemented in the chimpanzee dataset [23]. We chose to exclude these additional movements from the chimpanzee and domestic cat datasets to ensure the comparability of our observed configuration lists. For example, AU12+AU25+AU26+M55 would be reduced to AU12+AU25+AU26 (after omitting M55). Comparisons are being made to better understand whether certain facial muscle movements and facial configurations are underexplored in a specific species or if this applies to both chimpanzees and domesticated cats. We currently lack data on the frequency of head movements, eye movements, vocalizations, positional behaviors, manual gestures, and gross behaviors during bouts of facial signaling in domesticated cats, and the prevalence of asymmetrical facial muscle movements in chimpanzees. As a result, we are unable to determine how the reporting of these movements compares across chimpanzees and domesticated cats. After removing additional movements, chimpanzees had 24 facial muscle movements, and domesticated cats had 29.

The process of removing additional movements also reduced the number of morphologically distinct facial signals in chimpanzees from 357 to 66 and from 276 to 275 in domesticated cats [18, 23]. As a result, our "observed" configuration list for chimpanzees experienced a significant decrease in the number of morphologically distinct muscle movement combinations. After removing these extra movements, the remaining facial signals were condensed into configurations that were already included in the previously published dataset. Information on previously observed facial signals for chimpanzees and domesticated cats, along with a revised list of facial configurations used for our current study, can be found in the electronic supplement (**S2, S3 Tables in S1 File**, respectively).

All FACS coding was performed in ELAN AVFX using a custom annotation template [41]. Consistent with previous studies, we assessed FACS coding agreement using Wexler's ratio [15, 36, 42]. A Wexler's ratio of 0.70 is the minimum requirement for passing FACS certification tests and is considered 'good' agreement in research studies [42]. The last author (BNF) was certified in chimpFACS in 2016 and catFACS in 2021 and served as the primary FACS coder for both datasets. Assistance in assessing inter-rater agreement was provided by an undergraduate research assistant (Sarah Yadegari, SY) for chimpFACS coded data and by second author (LS) for catFACS coded data. SY obtained chimpFACS certification in 2019, and LS obtained catFACS certification in 2022. We assessed agreement using 10% of video footage with chimpanzees and domesticated cats, resulting in average Wexler's ratios of 0.750 and 0.756, respectively. BNF used chimpFACS and catFACS to code the remaining video footage and facial signaling observations after reaching good agreement.

## Possible combinations—Combinatorics in Python

To generate a list of all "possible" facial configurations for each species, we developed two combinatorial models for in Python: one for chimpanzees and one for domesticated cats. To provide a brief overview, each model generates a list of all possible facial configurations using

guidelines about the relationship between each facial muscle and movement. Guidelines regarding facial configurations were taken from the chimpFACS [5] and catFACS manuals [43]. In cases where guidelines were unclear or absent for facial muscle movements, we consulted the humanFACS [3, 44] manual for additional guidance, as long as movements were homologous across both species. For example, AU28 (lip suck) and AD32 (bite) are mentioned in both the chimpFACS [5] and humanFACS [3, 44]. We consulted the humanFACS [3, 44] manual to confirm if both these facial muscle movements can appear together, as it was not mentioned in the chimpFACS manual [5]. A list of combination guidelines used in our chimpanzee and domesticated cat combinatorial models can be accessed in the electronic supplement (**S2 File**).

Additionally, these models can cross-check previously documented facial configurations against those generated by the model using the FACS guidelines to assess coding accuracy (i.e., whether the combination is possible). All previously documented facial configurations that satisfied our FACS coding guidelines were appended to a new "observed" list (**S2 File**). Configurations that did not comply with our FACS coding guidelines were moved to a separate "flagged" category for manual assessment (**S2 File**). Facial configurations that were generated by our combinatorial models (based on FACS coding guidelines) but were not observed in previously published datasets were moved to their own "unobserved" lists (**S2 File**). Our combinatorial models and corresponding Python code are available in the electronic supplement (**S3** and **S4** **Files**).

## Data analysis

In addition to characterizing the outputs of our combinatorial models (i.e., our "possible" configuration lists) using descriptives, we also compared them to our "observed" configuration lists. Specifically, we compared the proportion of facial muscle movements observed across all facial configurations in our "observed" lists for each species (by taking the raw counts for each facial muscle movement and dividing them by the total number of observed facial configurations) to proportions of facial muscle movements generated by our combinatorial models (i.e., our "possible" configuration lists). Making comparisons between our configuration lists provides the opportunity to see whether some facial muscle movements are over-represented or under-represented in "observed" configuration lists.

## Results

Using the approaches described above, we identified 24 discrete facial muscle movements for chimpanzees and 29 for domesticated cats. In chimpanzees, these 24 discrete facial muscle movements were used to produce 66 unique facial signals (i.e., communicative facial configurations). In domesticated cats, 29 discrete facial muscle movements were used to produce 275 unique facial signals.

We also generated "possible" configuration lists for chimpanzees and domesticated cats using two Python combinatorial models. These models produced a list of all possible facial configurations (including both communicative and non-communicative) using coding guidelines outlined in the FACS manual. Using the same number of discrete facial muscle movements (N = 24 for chimpanzees and N = 29 for domesticated cats), our models generated a total of 238,079 unique facial configurations for chimpanzees and 1,062,719 for domesticated cats. These numbers include all "possible" facial configurations, including both "observed" and "unobserved" configurations. A detailed list of all unique facial configurations that were generated by our Python models and identified in previous studies can be found in the

electronic supplement (**S2 File**). Below, we analyze our "observed" configuration lists in relation to our "unobserved" configuration lists for each species using our Python models.

## Chimpanzees

Out of the 238,079 facial configurations identified for chimpanzees (in our "possible" list), 57 have been "observed" during bouts of communication in previous studies (0.024%) and 238,022 have yet to be observed (i.e., "unobserved" list; 99.976%). Of the 66 distinct facial signals found in chimpanzees, 9 were "flagged" (i.e., they were included in our "observed" list but were not found in our "possible" configuration list). We also examined differences in the number (and proportion) of discrete facial muscle movements identified in our "observed" configuration list and "unobserved" configuration list (**Table 1**). Of the 24 discrete facial muscle movements, 21 occurred more frequently in the "unobserved" configuration list than in the "observed" list.

## Domesticated cats

Out of the 1,062,719 facial configurations identified for domesticated cats (in our "possible" list), 272 have been observed during bouts of communication in previous studies (0.026%) and

**Table 1. A list of all facial muscle movements mentioned in the chimpFACS manual [5] and considered in our present study.** We include the raw count and proportion of facial muscle movements identified in previous studies (i.e., our "observed" configuration list), as well as the raw count and proportion of facial muscle movements that were in our "unobserved" configuration list (identified using our combinatorial model). The proportion of "observed" facial muscle movements was calculated by dividing the number of instances that each facial muscle movement was observed (in column 3) by the total number of facial configurations in our "observed" list (N = 57). The proportion of "unobserved" facial muscle movements was calculated by dividing the number of instances that each facial muscle was generated by our Python model (in column 5) by the total number of facial configurations in our "unobserved" list (N = 238,022). The last column displays the difference in proportion between observed and unobserved facial muscle movements (by subtracting column 4 from column 6).

| Code | FACS Description | No. Observed | Prop. Observed | No. Unobserved | Prop. Unobserved | Diff. |
|------|------------------|--------------|----------------|----------------|------------------|-------|
| AU1+2 | Inner & Outer Brow Raiser | 0 | 0.000% | 177,408 | 74.534% | -74.534% |
| AU6 | Cheek Raiser | 11 | 19.298% | 119,027 | 50.007% | -30.709% |
| AU43 | Eyes Closed | 8 | 14.035% | 79,353 | 33.339% | -19.304% |
| AU45 | Blink | 0 | 0.000% | 79,360 | 33.341% | -33.341% |
| AU9 | Nose Wrinkler | 10 | 17.544% | 119,026 | 50.006% | -32.462% |
| AU10 | Upper Lip Raiser | 21 | 36.842% | 119,020 | 50.004% | -13.162% |
| AU12 | Lip Corner Puller | 35 | 61.404% | 119,006 | 49.998% | 11.406% |
| AU16 | Lower Lip Depressor | 33 | 57.895% | 53,728 | 22.573% | 35.322% |
| AU17 | Chin Raiser | 2 | 3.509% | 7,678 | 3.226% | 0.283% |
| AU22 | Lip Funneler | 5 | 8.772% | 53,755 | 22.584% | -13.812% |
| AU24 | Lip Pressor | 1 | 1.754% | 6,143 | 2.581% | -0.827% |
| AU25 | Lips Part | 55 | 96.491% | 214,968 | 90.314% | 6.177% |
| AU26 | Jaw Drop | 31 | 54.386% | 130,529 | 54.839% | -0.453% |
| AU27 | Mouth Stretch | 17 | 29.825% | 73,712 | 30.969% | -1.144% |
| AU28 | Lip Suck | 0 | 0.000% | 52,224 | 21.941% | -21.941% |
| AD19 | Tongue Out | 7 | 12.281% | 98,297 | 41.297% | -29.016% |
| AD21 | Neck Tightener | 0 | 0.000% | 119,040 | 50.012% | -50.012% |
| AD29 | Jaw Thrust | 0 | 0.000% | 119,040 | 50.012% | -50.012% |
| AD30 | Jaw Sideways | 0 | 0.000% | 119,040 | 50.012% | -50.012% |
| AD32 | Lip Bite | 0 | 0.000% | 86,016 | 36.138% | -36.138% |
| AD33 | Cheek Blow | 0 | 0.000% | 119,040 | 50.012% | -50.012% |
| AD35 | Cheek Suck | 0 | 0.000% | 62,976 | 26.458% | -26.458% |
| AD37 | Lip Wipe | 0 | 0.000% | 98,304 | 41.300% | -41.300% |
| AD160 | Low Lip Relaxer | 6 | 10.526% | 53,754 | 22.584% | -12.058% |

**Table 2. A list of all facial muscle movements mentioned in the catFACS manual [14] and considered in our present study.** We include the raw count and proportion of facial muscle movements identified in previous studies (i.e., our "observed" configuration list), as well as the raw count and proportion of facial muscle movements that were in our "unobserved" configuration list (identified using our combinatorial model). The proportion of "observed" facial muscle movements was calculated by dividing the number of instances that each facial muscle movement was observed (in column 3) by the total number of facial configurations in our "observed" list (N = 272). The proportion of "unobserved" facial muscle movements was calculated by dividing the number of instances that each facial muscle was generated by our Python model (in column 5) by the total number of facial configurations in our "unobserved" list (N = 1,062,447). The last column displays the difference in proportion between observed and unobserved facial muscle movements (by subtracting column 4 from column 6).

| Code | FACS Description | No. Observed | Prop. Observed | No. Unobserved | Prop. Unobserved | Diff. |
|---|---|---|---|---|---|---|
| AU5 | Upper Lid Raiser | 72 | 26.471% | 212,468 | 19.998% | 6.473% |
| AU143 | Eyes Closed | 37 | 13.603% | 212,507 | 20.002% | -6.399% |
| AU145 | Blink | 3 | 1.103% | 212,541 | 20.005% | -18.902% |
| AU47 | Half Blink | 89 | 32.721% | 212,455 | 19.997% | 12.724% |
| AD48 | Third eyelid show | 0 | 0.000% | 531,360 | 50.013% | -50.013% |
| AD68 | Pupil dilator | 52 | 19.118% | 354,188 | 33.337% | -14.219% |
| AD69 | Pupil constrictor | 66 | 24.265% | 354,174 | 33.336% | -9.071% |
| AU109+110 | Nose Wrinkler | 40 | 14.706% | 531,320 | 50.009% | -35.303% |
| AU12 | Lip Corner Puller | 80 | 29.412% | 485,078 | 45.657% | -16.245% |
| AU116 | Lower Lip Depressor | 50 | 18.382% | 492,430 | 46.349% | -27.967% |
| AU17 | Chin Raiser | 0 | 0.000% | 25,920 | 2.440% | -2.440% |
| AU118 | Lip Puckerer | 1 | 0.368% | 354,239 | 33.342% | -32.974% |
| AU25 | Lips Part | 140 | 51.471% | 984,820 | 92.694% | -41.223% |
| AU26 | Jaw Drop | 127 | 46.691% | 492,353 | 46.341% | 0.350% |
| AU27 | Mouth Stretch | 13 | 4.779% | 466,547 | 43.912% | -39.133% |
| AD19 | Tongue Out | 4 | 1.471% | 311,036 | 29.275% | -27.804% |
| AD37 | Lip Wipe | 13 | 4.779% | 311,012 | 29.273% | -24.494% |
| AD137 | Nose Lick | 45 | 16.544% | 310,995 | 29.272% | -12.728% |
| AD190 | Tongue downwards | 7 | 2.574% | 311,033 | 29.275% | -26.701% |
| AU200 | Whiskers retractor | 22 | 8.088% | 354,218 | 33.340% | -25.252% |
| AU201 | Whiskers protractor | 20 | 7.353% | 354,220 | 33.340% | -25.987% |
| AU202 | Whiskers raiser | 16 | 5.882% | 531,344 | 50.011% | -44.129% |
| EAD101 | Ears Forward | 81 | 29.779% | 309,879 | 29.167% | 0.612% |
| EAD102 | Ear Adductor | 85 | 31.250% | 309,875 | 29.166% | 2.084% |
| EAD103 | Ear Flattener | 70 | 25.735% | 309,890 | 29.168% | -3.433% |
| EAD104 | Ear Rotator | 149 | 54.779% | 398,371 | 37.496% | 17.283% |
| EAD105 | Ears Downward | 41 | 15.074% | 265,639 | 25.003% | -9.929% |
| EAD106 | Ears Backwards | 7 | 2.574% | 132,833 | 12.503% | -9.929% |
| EAD107 | Ears constrictor | 6 | 2.206% | 177,114 | 16.670% | -14.464% |

1,062,447 have yet to be observed (i.e., "unobserved" list; 99.974%). Of the 275 distinct facial signals found in domesticated cats, 3 were "flagged" (i.e., they were included in our "observed" configuration list but were not found in our "possible" configuration list). We also examined differences in the number (and proportion) of discrete facial muscle movements identified in our "observed" configuration list and "unobserved" configuration list (**Table 2**). Of the 29 discrete facial muscle movements, 23 occurred more frequently in the "unobserved" configuration list than in the "observed" list.

## Discussion

Our current study aimed to examine the facial behavior of chimpanzees (*Pan troglodytes*) and domesticated cats (*Felis silvestris catus*) using Facial Action Coding Systems (FACS) and combinatorial techniques. First, we developed Python models capable of generating a list of all

possible facial configurations (i.e., muscle movement combinations) each mammal can produce using FACS coding and their associated guidelines [5, 14]. Then, we compared the list of facial configurations generated by our Python models with the lists of previously identified facial signals for each mammal [18, 23].

Our results suggest that chimpanzees and domesticated cats may be capable of producing a wide variety of facial configurations, many of which have not been observed or documented. It is plausible that some unobserved facial configurations are communicative (i.e., used as facial signals). Compared to hylobatids, whose facial signals have been documented across multiple studies involving many individuals and hours of video footage [17, 18, 24, 25], the facial signaling repertoires of chimpanzees and domesticated cats may be limited due to a lack of video footage, research with additional populations, and/or small sample sizes (of individuals). Although chimpanzees and domesticated cats are widely studied with FACS, there are few studies that publish comprehensive lists of facial signals. Additional studies with larger and diverse populations are necessary to fully assess the communicative repertoires of both mammals.

It is important to note that in order to determine whether a facial configuration is communicative, the behavioral responses of recipients must be evaluated. If signals help to elicit behavioral responses in recipients that have positive fitness consequences for both signalers and recipients [40], then there should be observable behavioral responses from recipients. This may also involve a clear attempt to elicit a behavioral response from the signaler's perspective, with evidence of intentionality and/or goal-directed behavior [25, 45]. It is only by coding the behavioral responses of both signalers and recipients that we can determine which configurations generated by our Python models are communicative or non-communicative. Increasing sample sizes and hours of video footage may reveal additional combinations that are used as facial signals, but it is also possible that our "observed" datasets represent the full communicative potential of each species.

By comparing previously observed facial signals with unobserved facial configurations generated through combinatorial approaches, we can help researchers decide whether to allocate additional time and resources to study the facial signaling repertoires of chimpanzees and domesticated cats in the future. Researchers who invest a lot of time and resources in collecting and coding facial signaling data using FACS may face high risks and limited rewards if they are solely focused on documenting facial signaling repertoires. This is especially true if our "observed" configuration list represents the full communicative potential of each species. For instance, previous studies have estimated that coding 1 minute of video footage with FACS takes approximately 1–2 hours [46, 47]. This estimate does not account for the roughly 100 hours needed to attain proficiency and certification in using FACS [46, 47] or the amount of time needed to collect video footage for coding and assessing inter-observer reliability. Certification in FACS assesses expertise, but inter-observer reliability is essential to evaluate ongoing consistency in behavioral coding, as proficiency may diminish over time. Conducting inter-observer reliability can also be a time-intensive endeavor. But future studies do not have to solely focus on documenting each species' communicative potential. Instead, their findings can be indirectly used to achieve this purpose. There are various research questions that FACS-based approaches can address, such as detecting instances of rapid facial mimicry [48] or documenting the effects of multimodal communication on facial signaling form and function [49]. Information on facial signaling repertoires can be used to detect instances of rapid facial mimicry and assess the role of multimodal communication, while also contributing to a comprehensive list of communicative facial configurations for chimpanzees and domesticated cats.

Because there are more documented AUs/ADs in the catFACS manual compared to the chimpFACS manual, our model generated a greater number of facial configurations for domesticated cats. Domesticated cats, for example, exhibit a wide range of ear movements that have not been described in the chimpFACS [5]. Furthermore, additional movements associated with the pupils, whiskers, and tongue have been documented in the catFACS [43] but not in the chimpFACS [5, 36]. However, our estimates regarding the number of possible facial configurations for each species may change when including additional facial action descriptors (such eye movements, head positions, and head movements). When these additional facial muscle movements and descriptors are considered, the number of morphologically distinct facial signals in chimpanzees, for example, increases from 66 to 357 [18]. However, future studies should assess whether these additional facial action descriptors are communicative before considering their inclusion into the observed list of facial configurations. Greater facial mobility does not necessarily lead to larger or more complex facial signaling repertoires, however, as the communicative potential of each species is largely shaped by their socio-ecologies [18]. But given the large-scale social living of chimpanzees [18] and the social flexibility of domesticated cats [50] it is more plausible that a greater number of morphologically distinct facial signals will be identified for both mammals. Taken together, our findings suggest that studying the communication abilities of chimpanzees and domesticated cats is a worthwhile endeavor that can be achieved indirectly through collaborative research partnerships.

In addition to comparing the number of previously documented facial configurations to the facial configurations generated through our combinatorial models, we also examined the proportion of discrete facial muscle movements observed across configurations. Comparing proportions of facial muscle movements can help identify whether some facial muscle movements are at greater risk of being overlooked during the coding process, which may lead to underreporting. We have used our unobserved configuration lists as a reference point, as they may represent the full communicative potential of each species (although additional research is needed to verify this idea). In our analysis, we found that some facial muscle movements were identified at similar rates in our "observed" and "unobserved" configuration lists. Examples include AUs/ADs that have less than a 1% difference in documented proportions, such as AU17 (chin raiser) and AU26 (jaw drop) in chimpanzees and AU26 (jaw drop) and EAD101 (ears forward) in domesticated cats. However, many of the facial muscle movements included in our study appear more frequently in our unobserved configuration lists than in our observed configuration lists, which could suggest underreporting.

If greater differences in proportions are indeed the result of underreporting, then there are two plausible explanations for why this underreporting is occurring. First, some facial muscle movements are subtle and difficult to identify, such as nose wrinkling (AU9 in chimpanzees and AU109+110 in domesticated cats; [5, 14]. We indeed found that these muscle movements were more prevalent in our unobserved configuration lists compared to our observed lists. AU9 and AU109+110 were more common in our unobserved configuration lists than in our observed lists. It can be challenging to identify nose wrinkling movements, especially when they are not very pronounced [44]. Nose wrinkling movements are crucial in the production of non-affiliative facial signals, as they have been previously documented in the submissive and aggressive facial displays of chimpanzees [36] and domesticated cats [22]. These studies further contribute to the possibility that these facial muscle movements are being underreported in observed configuration lists. Second, there might be cases where the FACS coder(s) have overlooked certain facial muscle movements due to the presence of several other movements happening at the same time. For example, the we observed a greater proportion of tongue displays (AD19 in both chimpanzees and domesticated cats) in our unobserved configuration lists when compared to our observed configuration lists. In order to stick the

tongue out of the mouth, other facial muscle movements such as lip parting (AU25) and jaw drop/stretch (AU26/27) must occur.

Our example of potential underreporting of AD19 also leads us to one of the strengths of our study: our generated lists of facial configurations can improve coding accuracy. Our configurations lists can highlight underreported facial muscle movements and also evaluate combinations of movements according to FACS guidelines. Our combinatorial model in Python was able to detect 57 out of the 66 morphologically distinct facial signals found in our "observed" configuration list for chimpanzees. Similarly, it detected 272 out of the 275 morphologically distinct facial signals found in "observed" configuration list for domesticated cats. There are three possible explanations as to why our combinatorial model flagged 9 chimpanzee facial signals in our "observed" configuration list. First, some of these facial signals may have been produced on only one side of the face, resulting in asymmetry. The second possibility is that there could have been multiple signal peaks due to blended effects. This means that multiple types of communicative facial configurations are rapidly combined together, making it difficult to discern which apex to focus on for coding. Finally, these flagged facial signals may be the result of manual coding errors. The three facial signals that were flagged by our cat combinatorial model appear to reflect coding errors associated with visibility issues. In two cases, there appeared to be difficulty distinguishing between ear rotating and narrowing movements (EAD104 versus EAD106). In the last case, there was no direct observation of lip parting movement (AU25), but evidence of tongue protrusion (AD37) was present.

Our combinatorial models still reveal a high level of accuracy for our two previously published FACS-coded datasets. Most of the facial configurations observed in our lists for chimpanzees (86.364%) and domesticated cats (98.910%) were not flagged, meaning that they conformed to the FACS guidelines. Manual coders can maintain a high level of accuracy by frequently referencing the FACS for guidance and renewing certifications when significant time has passed. However, these techniques may not be as easily applied when coding automation is being used. In recent years, there has been rapid development of automated facial muscle movement detection systems, most notably using landmark features [51]. We suggest utilizing these automated techniques in conjunction with our combinatorial models to provide an additional level of assistance in accurately identifying facial muscle movements during communication. Many of these automated approaches have been developed with inspiration from FACS, making our combinatorial approaches compatible. For example, landmark detection systems for domesticated cats have been developed using inspiration from FACS-based coding techniques [50, 52–54]. These landmark detection systems achieve a high level of accuracy when using high-quality video footage gathered in controlled environments [52]. Since a large portion of animalFACS studies are conducted in captive environments (including those with chimpanzees and domesticated cats; [18, 22, 23, 30, 43, 55, 56], these landmark detection techniques can be highly beneficial and dependable. Configuration data generated from our combinatorial models can be used to assess the accuracy of landmark detection outputs by determining the feasible combinations of facial muscle movements.

However, it is possible that differences in the proportion of identified facial muscle movements in our "observed" and "unobserved" configuration lists signify differences in their social function rather than underreporting due to coding error. Previous studies with chimpanzees, for example, found that certain facial muscle movements are associated with particular types of facial signals that often have strong associations with a given context [36]. For example, AU10 (Upper Lip Raiser) is typically associated with the production of scream faces, and seldom appears in other types of communicative displays (like play faces and pouts; [36]). In contrast, other facial muscle movements (such as AU12) appear across multiple facial signal types (such as bared-teeth displays, play faces, and scream faces; [36]). In our current research, we

noticed that AU10 appeared more frequently in unobserved facial configurations than in observed ones, but the opposite pattern was observed for AU12. This pattern could have been observed because AU12 has multiple communicative functions, while AU10 is limited in its communicative functioning. Additional research should focus on discerning the communicative function of discrete facial muscle movements, in addition to the function of facial configurations, to explore this possibility further.

While our study aims to shed light on the number of morphologically distinct facial configurations that chimpanzees and domesticated cats potentially produce, we also believe that our generated list of facial muscle movement combinations can be a valuable resource for researchers who are interested in applying similar approaches to other mammals. FACS can also be used to document the facial cues/signals repertoires of monkeys [8–12], dogs [13], and horses [15]. When datasets containing information on their facial muscle movement combinations become available, our guidelines can be modified and applied to these species. Our study results can also aid researchers in assessing FACS coding accuracy by comparing their datasets to a pre-existing facial signal database created using configuration guidelines. Using combinatorial techniques, we can compare possible facial muscle movements with those previously observed and provide recommendations for coding procedures for subtle movements. It is essential to point out that our current study does not discern whether variations in subtle movements occur due to differences in usage or visibility. Instead, the differences in the documentation of subtle facial muscle movements could indicate that more research is necessary to distinguish where these differences originate from.

To conclude, our study reveals that chimpanzees and domesticated cats may communicate using a greater number of facial signals than previously documented, which requires further research. Our research has also identified new opportunities for investigating the communicative abilities of other mammal species beyond chimpanzees and domesticated cats. Finally, there is potential for using advanced techniques to improve the accuracy of manual coding or automate such processes.

## Supporting information

**S1 Text. Datasets (for chimpanzees and domesticated cats) and Python code can be found in the online electronic supplement (S1–S4 Files).** Please see below for more information about each file.
(DOCX)

**S1 File. Supplementary Tables and Information.** This document contains supplementary tables and detailed information about the Excel workbook where our raw data is stored (S2 File, below).
(DOCX)

**S2 File. Raw data.** This Excel workbook contains all the data used for our current study. Descriptions for each Excel sheet can be found in the S1 File.
(XLSX)

**S3 File. Python code for chimpanzees.** This PDF contains information about the Python code used to generate our list of possible combinations for chimpanzees.
(PDF)

**S4 File. Python code for cats.** This PDF contains information about the Python code used to generate our list of possible combinations for domesticated cats.
(PDF)

## Acknowledgments

We would like to express our gratitude to the directors, staff, and volunteers at the CatCafé Lounge, Stray Cat Alliance, and the Los Angeles Zoo and Botanical Gardens for allowing us to access their animals. A special thank you goes to Sarah Yadegari for assisting with inter-observer reliability for chimpFACS coded data. We are also thankful to Dr. Matthew Campbell from California State University Channel Islands for providing feedback on the framing of our manuscript, as well as Dr. Marcus Birkenkrahe from Lyon College for the feedback on the organization of our Python code. Additionally, we appreciate the assistance of the Animal-FACS team in chimpFACS and catFACS certification.

## Author Contributions

**Conceptualization:** Aisha Mahmoud, Lauren Scott, Brittany N. Florkiewicz.

**Data curation:** Aisha Mahmoud, Lauren Scott, Brittany N. Florkiewicz.

**Formal analysis:** Aisha Mahmoud, Brittany N. Florkiewicz.

**Investigation:** Aisha Mahmoud, Lauren Scott, Brittany N. Florkiewicz.

**Methodology:** Aisha Mahmoud, Lauren Scott, Brittany N. Florkiewicz.

**Project administration:** Brittany N. Florkiewicz.

**Resources:** Aisha Mahmoud, Lauren Scott, Brittany N. Florkiewicz.

**Software:** Aisha Mahmoud.

**Supervision:** Brittany N. Florkiewicz.

**Validation:** Aisha Mahmoud, Lauren Scott, Brittany N. Florkiewicz.

**Visualization:** Aisha Mahmoud, Brittany N. Florkiewicz.

**Writing – original draft:** Aisha Mahmoud, Lauren Scott, Brittany N. Florkiewicz.

**Writing – review & editing:** Aisha Mahmoud, Lauren Scott, Brittany N. Florkiewicz.

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
