## [Decision Letter · Decision Letter 0]

21 Oct 2024

PONE-D-24-42931Examining Mammalian Facial Behavior using Facial Action Coding Systems (FACS) and CombinatoricsPLOS ONE

Dear Dr. Florkiewicz,

Thank you for submitting your manuscript to PLOS ONE. After careful consideration, we feel that it has merit but does not fully meet PLOS ONE’s publication criteria as it currently stands. Therefore, we invite you to submit a revised version of the manuscript that addresses the points raised during the review process.

We look forward to receiving your revised manuscript.

Kind regards,

Tomoyoshi Komiyama, Ph.D

Academic Editor

PLOS ONE

Journal requirements: When submitting your revision, we need you to address these additional requirements. 1. Please ensure that your manuscript meets PLOS ONE's style requirements, including those for file naming. The PLOS ONE style templates can be found at https://journals.plos.org/plosone/s/file?id=wjVg/PLOSOne_formatting_sample_main_body.pdf and https://journals.plos.org/plosone/s/file?id=ba62/PLOSOne_formatting_sample_title_authors_affiliations.pdf 2. PLOS requires an ORCID iD for the corresponding author in Editorial Manager on papers submitted after December 6th, 2016. Please ensure that you have an ORCID iD and that it is validated in Editorial Manager. To do this, go to ‘Update my Information’ (in the upper left-hand corner of the main menu), and click on the Fetch/Validate link next to the ORCID field. This will take you to the ORCID site and allow you to create a new iD or authenticate a pre-existing iD in Editorial Manager. 3. Please note that PLOS ONE has specific guidelines on code sharing for submissions in which author-generated code underpins the findings in the manuscript. In these cases, we expect all author-generated code to be made available without restrictions upon publication of the work. Please review our guidelines at https://journals.plos.org/plosone/s/materials-and-software-sharing#loc-sharing-code and ensure that your code is shared in a way that follows best practice and facilitates reproducibility and reuse. 4. Please include captions for your Supporting Information files at the end of your manuscript, and update any in-text citations to match accordingly. Please see our Supporting Information guidelines for more information: http://journals.plos.org/plosone/s/supporting-information. 

Additional Editor Comments:

Dear authors,

Thank you for submitting your revised manuscript.

Your research investigated facial cues/signals in chimpanzees and domesticated cats and suggested further research may discover additional undocumented it.

These findings could have major implications for mammal communicative research and may assist researchers in evaluating FACS coding accuracy.

It was much easier to understand than the original manuscript.

However, two reviewers had additional comments.

Please answer these questions as listed below.

I believe this manuscript will satiate the reader's interest.

Tomoyoshi Komiyama

Reviewers' comments:

Reviewer's Responses to Questions

**Comments to the Author**

1. Is the manuscript technically sound, and do the data support the conclusions?

Reviewer #1: Partly

Reviewer #2: Yes

2. Has the statistical analysis been performed appropriately and rigorously? 

Reviewer #1: N/A

Reviewer #2: N/A

3. Have the authors made all data underlying the findings in their manuscript fully available?

Reviewer #1: Yes

Reviewer #2: Yes

4. Is the manuscript presented in an intelligible fashion and written in standard English?

Reviewer #1: Yes

Reviewer #2: Yes

5. Review Comments to the Author

Reviewer #1: I would first like to commend the authors on so diligently integrating the reviewer’s comments into their revised manuscript. Many of the points that were raised by reviewers in the previous version have been addressed. However, while I acknowledge the immense amount of effort and time that went into the coding of the videos and the crafting of the Python code, I remain at present not entirely certain I have been convinced as to one of the major proposed benefits of this method. The authors very correctly describe how time consuming it can be to code behavior using FACs, and that there may be diminishing returns for such labor intensive coding if the only goal is to document a facial signaling repertoire. However, the solution that the authors present is to give researchers a tool that will enable them to calculate the theoretical maximum number of potentially communicative facial configurations an animal can produce. If I were studying an animal in which we had observed, say, 400 observed combinations, but when I ran your model I was told that there are >1,000,000 possible combinations I have not observed, I would either conclude that 1. I have a lot more work to do, and I would devote significantly more time to investigating these unobserved but possible combinations or 2. That the vast majority of these are only possible in theory and not in practice.

To use another illustration, say you were told that some language could have words between 3 to 15 letters long, with 26 letters in them. Your script could generate presumably millions of possible words. Many may be real words. But the vast majority are probably not words at all (think of how many grammatical nonsense words there are in English, for instance. Help is a word, delp is not). Knowing that there are many hundreds of thousands or millions of possible words will get me no closer to knowing the actual lexicon of a human language, or non human facial signaling system, nor will it help me to decide when to stop looking for more.

The method I described in my last review in which you look at the discovery curve for new signals after certain amounts of effort hours would be the only way in which I could evaluate whether I have reached a point of diminishing returns. As a result, at least half of your justification I am having a difficult time understanding.

That being said, I do absolutely agree that your script could be used for validation, and I especially liked your addition of this validation being paired with automated coding methods which are far less intelligent than actual humans. If anything, I would downplay the argument about this being a tool to help researchers decide when to stop sampling, and focus on this element.

Finally, forgive me but I am still not certain I understand why you did not use the fill chimpanzee facial action unit dataset. The authors say it is to ensure comparability of results between chimpanzees and cats, but you are not comparing chimpanzees to cats, you are comparing observed chimpanzees to theoretically possible chimpanzees, correct? In your discussion, you mention that the number of possible facial action units in chimpanzees is larger, and including this full dataset could increase the possible number of combinations considerably. But then say these would need to be first evaluated to assess if they are communicative. This is true for every combination and facial action unit you observed as far as I understand. So I personally think you do not need to reduce the chimpanzee dataset to facilitate a comparison you do not conduct.

I apologize for continuing to be critical. Understanding nonhuman communication is a major focus of my work, and I appreciate the difficulties that come with trying to figure out how to define an animal’s repertoire. I hope these comments can be helpful in guiding your restructuring.

Reviewer #2: General comment:

I would like to thank the authors for carefully considering and answering my and the other reviewers’ comments on the previous version of the manuscript. While I am generally satisfied with the revisions, my previous main objection remains. I am not convinced that comparing the proportion of observed facial configurations with the total number of possible facial configurations is the way forward to make decisions about data collection and coding effort, or the assess the facial mobility and communication potential of a given species. Existing methods such as passing the FACS certification, using cumulative frequencies of facial combinations (similar to the species richness curve mentioned by reviewer 2) would achieve the same goals. These methods are already commonly used in communication research.

Detailed comments:

47 This is not entirely accurate. In Clark et al. (2020) FACS was used to measure morphological differences between facial configurations. Then, statistical methods were used to assess how these differences were associated to different interaction outcomes.

60 Which three mammals does this refer to? There is a range of species which were studied in the references listed above – more than three.

63-64 That is true, but it doesn't seem that the method you propose is capable to considering graded and blended signals either. It doesn’t seem fair to highlight this as a limitation to justify your approach when it suffers from the same issue.

107-109 As mentioned above, there is already such a method, based on the number of unique configurations observed over time. Optimal observation effort can be identified when the curved reach a plateau. See Hobaiter, C., & Byrne, R. W. (2011). The gestural repertoire of the wild chimpanzee. Animal cognition, 14, 745-767 for example.

145 I would use the past tense (compiled) since the study has already been conducted.

291-301 A lot, if not all the information presented here has already been presented in the methods. It only needs to appear once.

6. PLOS authors have the option to publish the peer review history of their article (what does this mean?). If published, this will include your full peer review and any attached files.

Reviewer #1: **Yes: **Severine Hex

Reviewer #2: No

---

## [Author Response · Author response to Decision Letter 0]

1 Nov 2024

Formatting Requirements

• We have revised the names of our supplementary files (and how they are referenced in the manuscript text; see the Supporting Information section).

We have made some minor formatting changes to ensure our manuscript document abides by PLOS One’s formatting guidelines. These include the following changes:

• Using level 1 headings for all major sections (in 18-point font)

• Using level 2 headings for all subsections (16-point font)

• Including a section on Supporting Information (at the very end of the manuscript, after the reference section)

• All authors now have registered ORCID’s. However, when the first author (BF) goes to add in her ORCID, the following error message appears: “An identical, Validated ORCID already exists in the database. An ORCID may be linked to only one record in the database, and so you may have registered another user record in the past and retrieved/validated your ORCID for use with that.” So we are providing ORCID’s below:

o Brittany Florkiewicz: 0000-0001-7500-2455

o Aisha Mahmoud: 0009-0002-0235-8885

o Lauren Scott: 0000-0002-5423-0042

3. Please note that PLOS ONE has specific guidelines on code sharing for submissions in which author-generated code underpins the findings in the manuscript. In these cases, we expect all author-generated code to be made available without restrictions upon publication of the work. 

• Information about how to access our Python code can be found in the section entitled “Supporting Information” at the end of the manuscript.

4. Please include captions for your Supporting Information files at the end of your manuscript, and update any in-text citations to match accordingly. 

• We have added a section for Supporting Information files at the end of our manuscript.

Reviewer #1

I would first like to commend the authors on so diligently integrating the reviewer’s comments into their revised manuscript. Many of the points that were raised by reviewers in the previous version have been addressed. However, while I acknowledge the immense amount of effort and time that went into the coding of the videos and the crafting of the Python code, I remain at present not entirely certain I have been convinced as to one of the major proposed benefits of this method. The authors very correctly describe how time consuming it can be to code behavior using FACs, and that there may be diminishing returns for such labor intensive coding if the only goal is to document a facial signaling repertoire. However, the solution that the authors present is to give researchers a tool that will enable them to calculate the theoretical maximum number of potentially communicative facial configurations an animal can produce. If I were studying an animal in which we had observed, say, 400 observed combinations, but when I ran your model I was told that there are >1,000,000 possible combinations I have not observed, I would either conclude that 1. I have a lot more work to do, and I would devote significantly more time to investigating these unobserved but possible combinations or 2. That the vast majority of these are only possible in theory and not in practice. To use another illustration, say you were told that some language could have words between 3 to 15 letters long, with 26 letters in them. Your script could generate presumably millions of possible words. Many may be real words. But the vast majority are probably not words at all (think of how many grammatical nonsense words there are in English, for instance. Help is a word, delp is not). Knowing that there are many hundreds of thousands or millions of possible words will get me no closer to knowing the actual lexicon of a human language, or non human facial signaling system, nor will it help me to decide when to stop looking for more. 

Thank you for your feedback! It's crucial to address this point, as we don't want researchers to feel discouraged by the study's results. We've included the following information in the "Current Study" section to better highlight our study's objectives: “The results of our study are not meant to discourage researchers from conducting additional studies, even if millions of facial configurations are identified. Rather, we hope that our study can help researchers identify underexplored facial muscle movements and configurations, offering deeper insights into a species' socio-ecology.” (Lines 172-176).

We do acknowledge that not all facial configurations generated by our models will be communicative, as indicated in lines 168-172 of the Current Study section: “Although we may not be able to ascertain the communicative function of every facial configuration without additional research, compiling a thorough list of all possible facial configurations can still help in assessing the facial mobility and communication potential in mammals”. 

This point is further addressed in our discussion section (Lines 405-415): “It is important to note that in order to determine whether a facial configuration is communicative, the behavioral responses of recipients must be evaluated. If signals help to elicit behavioral responses in recipients that have positive fitness consequences for both signalers and recipients (Laidre & Johnstone, 2013), then there should be observable behavioral responses from recipients. This may also involve a clear attempt to elicit a behavioral response from the signaler's perspective, with evidence of intentionality and/or goal-directed behavior (Scheider et al., 2014; Byrne et al., 2017). It is only by coding the behavioral responses of both signalers and recipients that we can determine which configurations generated by our Python models are communicative or non-communicative.”

The method I described in my last review in which you look at the discovery curve for new signals after certain amounts of effort hours would be the only way in which I could evaluate whether I have reached a point of diminishing returns. As a result, at least half of your justification I am having a difficult time understanding. That being said, I do absolutely agree that your script could be used for validation, and I especially liked your addition of this validation being paired with automated coding methods which are far less intelligent than actual humans. If anything, I would downplay the argument about this being a tool to help researchers decide when to stop sampling, and focus on this element.

Another reviewer brought a similar idea (involving cumulative repertoire plots) to our attention, which we have added into our manuscript: “For this reason, it is crucial to have a method for assessing whether additional data collection and coding efforts will yield new facial signals or if a limit has been reached. Previous studies have identified communicative plateaus by plotting signaling time (in hours) against the cumulative repertoire size of individuals and species. For instance, studies with chimpanzees (Pan troglodytes) have revealed that after 15 hours of active signaling, there are minimal alterations in the gestural repertoire sizes of observed individuals (Hobaiter & Byrne, 2011)” (Lines 109-115).

In addition to including the example of the discovery curve in our previous round of edits, we have also added additional information about the strengths and limitations of our new approach, as well as how we envision our approach complementing techniques involving cumulative repertoire plots and discovery curves: “Cumulative repertoire plots (Hobaiter & Byrne, 2011) and discovery curves (Batista et al., 2021) provide one possible avenue for evaluating facial signal sampling efforts and repertoire thresholds. However, it's important to note that these approaches also have their limitations. Cumulative repertoire plots only display collective values over time, making it challenging to identify sudden fluctuations that may be due to other sampling factors and obstacles. Identifying and addressing these sampling factors and obstacles could result in a greater variety of facial signals than previously reported. Discovery curves are subject to large margins of error unless the inventory of signals is nearly complete (Bebber et al., 2007). Furthermore, discovery curves are prone to underestimation and false plateaus (Bebber et al., 2007), which could negatively impact the documentation of facial signaling repertoires. In order to improve the accuracy of cumulative repertoire plots and discovery curves, it is important to use an additional approach that can identify the potential maximum inventory of facial signals and detect any areas of possible sampling bias, such as coding bias.” (Lines 135-147).

Finally, forgive me but I am still not certain I understand why you did not use the fill chimpanzee facial action unit dataset. The authors say it is to ensure comparability of results between chimpanzees and cats, but you are not comparing chimpanzees to cats, you are comparing observed chimpanzees to theoretically possible chimpanzees, correct? In your discussion, you mention that the number of possible facial action units in chimpanzees is larger, and including this full dataset could increase the possible number of combinations considerably. But then say these would need to be first evaluated to assess if they are communicative. This is true for every combination and facial action unit you observed as far as I understand. So I personally think you do not need to reduce the chimpanzee dataset to facilitate a comparison you do not conduct.

Thank you for bringing this to our attention! We have revised the paragraph to provide more details and clarify the process of reducing our chimpanzee dataset and the nature of the comparisons being made: “There are different types of facial muscle movements recognized in the FACS. For our current study, we focus only on Action Units (AUs) and Action Descriptors (ADs). Our chimpanzee dataset contained additional movements such as head (M51-M55) and eye (M69) movements, vocalizations (AD50), positional behaviors (S101), manual gestures (S100), and gross behaviors (G84 & G85; Florkiewicz et al., 2023), but these were not coded for domesticated cats (Scott & Florkiewicz et al., 2023). One instance of asymmetrical ear movement was identified in the domestic cat dataset (Scott & Florkiewicz et al., 2023), but asymmetrical coding was not implemented in the chimpanzee dataset (Scott & Florkiewicz et al., 2023). We chose to exclude these additional movements from the chimpanzee and domestic cat datasets to ensure the comparability of our observed configuration lists. For example, AU12+AU25+AU26+M55 would be reduced to AU12+AU25+AU26 (after omitting M55). Comparisons are being made to better understand whether certain facial muscle movements and facial configurations are underexplored in a specific species or if this applies to both chimpanzees and domesticated cats. We currently lack data on the frequency of head movements, eye movements, vocalizations, positional behaviors, manual gestures, and gross behaviors during bouts of facial signaling in domesticated cats, and the prevalence of asymmetrical facial muscle movements in chimpanzees. As a result, we are unable to determine how the reporting of these movements compares across chimpanzees and domesticated cats. After removing additional movements, chimpanzees had 24 facial muscle movements, and domesticated cats had 29”. (Lines 242-261).

I apologize for continuing to be critical. Understanding nonhuman communication is a major focus of my work, and I appreciate the difficulties that come with trying to figure out how to define an animal’s repertoire. I hope these comments can be helpful in guiding your restructuring.

We appreciate all of your feedback on our manuscript. Thank you for helping us improve our research article! 

Reviewer #2 

I would like to thank the authors for carefully considering and answering my and the other reviewers’ comments on the previous version of the manuscript. While I am generally satisfied with the revisions, my previous main objection remains. I am not convinced that comparing the proportion of observed facial configurations with the total number of possible facial configurations is the way forward to make decisions about data collection and coding effort, or the assess the facial mobility and communication potential of a given species. 

Existing methods such as passing the FACS certification, using cumulative frequencies of facial combinations (similar to the species richness curve mentioned by reviewer 2) would achieve the same goals. These methods are already commonly used in communication research.

Thank you for your additional feedback and suggestions! We have added some additional information to address some of the points you raised:

To address your point about cumulative frequencies, we have added the following pieces of information to our manuscript: 

 “For this reason, it is crucial to have a method for assessing whether additional data collection and coding efforts will yield new facial signals or if a limit has been reached. Previous studies have identified communicative plateaus by plotting signaling time (in hours) against the cumulative repertoire size of individuals and species. For instance, studies with chimpanzees (Pan troglodytes) have revealed that after 15 hours of active signaling, there are minimal alterations in the gestural repertoire sizes of observed individuals (Hobaiter & Byrne, 2011)”. (Lines 109-115).

In addition to including the example of the discovery curve in our previous round of edits, we have also added additional information about the strengths and limitations of our new approach, as well as how we envision our approach complementing techniques involving cumulative repertoire plots and discovery curves: “Cumulative repertoire plots (Hobaiter & Byrne, 2011) and discovery curves (Batista et al., 2021) provide one possible avenue for evaluating facial signal sampling efforts and repertoire thresholds. However, it's important to note that these approaches also have their limitations. Cumulative repertoire plots only display collective values over time, making it challenging to identify sudden fluctuations that may be due to other sampling factors and obstacles. Identifying and addressing these sampling factors and obstacles could result in a greater variety of facial signals than previously reported. Discovery curves are subject to large margins of error unless the inventory of signals is nearly complete (Bebber et al., 2007). Furthermore, discovery curves are prone to underestimation and false plateaus (Bebber et al., 2007), which could negatively impact the documentation of facial signaling repertoires. In order to improve the accuracy of cumulative repertoire plots and discovery curves, it is important to use an additional approach that can identify the potential maximum inventory of facial signals and detect any areas of possible sampling bias, such as coding bias.” (Lines 135-147).

We have added the following piece of information to address the role of FACS certification in facial configuration coding efforts: “Certification in FACS assesses expertise, but inter-observer reliability is essential to evaluate ongoing consistency in behavioral coding, as proficiency may diminish over time. Conducting inter-observer reliability can also be a time-intensive endeavor.” (Lines 427-430).

We do acknowledge that not all facial configurations generated by our models will be communicative, as indicated in lines 168-172 of the Current Study section: “Although we may not be able to ascertain the communicative function of every facial configuration without additional research, 

---

## [Decision Letter · Decision Letter 1]

19 Nov 2024

Examining Mammalian Facial Behavior using Facial Action Coding Systems (FACS) and Combinatorics

PONE-D-24-42931R1

Dear Dr. Florkiewicz,

We’re pleased to inform you that your manuscript has been judged scientifically suitable for publication and will be formally accepted for publication once it meets all outstanding technical requirements.

Kind regards,

Tomoyoshi Komiyama, Ph.D

Academic Editor

PLOS ONE

Additional Editor Comments (optional):

Dear authors,

Thank you for submitting your revised manuscript.

It was much easier to understand than the original manuscript.

I am satisfied with the responses and the edits, so I am happy to accept your study.

You have satisfactorily addressed the comments from the two reviewers.

Therefore, I have no further suggestions.

I believe this manuscript will satiate the reader's interest.

Tomoyoshi Komiyama

Reviewers' comments:

Reviewer's Responses to Questions

**Comments to the Author**

1. If the authors have adequately addressed your comments raised in a previous round of review and you feel that this manuscript is now acceptable for publication, you may indicate that here to bypass the “Comments to the Author” section, enter your conflict of interest statement in the “Confidential to Editor” section, and submit your "Accept" recommendation.

Reviewer #1: All comments have been addressed

2. Is the manuscript technically sound, and do the data support the conclusions?

Reviewer #1: Yes

3. Has the statistical analysis been performed appropriately and rigorously? 

Reviewer #1: Yes

4. Have the authors made all data underlying the findings in their manuscript fully available?

Reviewer #1: Yes

5. Is the manuscript presented in an intelligible fashion and written in standard English?

Reviewer #1: Yes

6. Review Comments to the Author

Reviewer #1: Thank you for again taking the time to integrate the additional comments from myself and the other reviewer. The clarifications and qualifications that were provided in text addressed my remaining concerns satisfactorily. I look forward to seeing this paper in print.

7. PLOS authors have the option to publish the peer review history of their article (what does this mean?). If published, this will include your full peer review and any attached files.

Reviewer #1: **Yes: **Severine Hex

---

## [Editor Report · Acceptance letter]

21 Nov 2024

PONE-D-24-42931R1 

PLOS ONE

Dear Dr. Florkiewicz, 

I'm pleased to inform you that your manuscript has been deemed suitable for publication in PLOS ONE. Congratulations! Your manuscript is now being handed over to our production team.

Kind regards, 

on behalf of

Dr. Tomoyoshi Komiyama 

Academic Editor

PLOS ONE